# Effectiveness of Calcium Phosphate Desensitising Agents in Dental Hypersensitivity Over 24 Weeks of Clinical Evaluation

**DOI:** 10.3390/nano9121748

**Published:** 2019-12-09

**Authors:** Paolo Usai, Vincenzo Campanella, Giovanni Sotgiu, Giovanni Spano, Roberto Pinna, Stefano Eramo, Laura Saderi, Franklin Garcia-Godoy, Giacomo Derchi, Giorgio Mastandrea, Egle Milia

**Affiliations:** 1Department of Biomedical Sciences, University of Sassari, 07100 Sassari, Italy; dottpaolousai@gmail.com (P.U.); caesareus83@yahoo.it (R.P.); 30050744@studenti.uniss.it (G.M.); 2Department of Clinical and Translational Medicine, Tor Vergata University of Rome, 00133 Rome, Italy; 3Department of Medical, Surgical and Experimental Sciences, University of Sassari, 07100 Sassari, Italy; lsaderi@uniss.it; 4Dental Unit, Azienda Ospedaliero-Universitaria, 07100 Sassari, Italy; giovanni.spano@aousassari.it; 5Department of Surgical and Biomedical Sciences, University of Perugia, S. Andrea delle Fratte, 06156 Perugia, Italy; stefano.eramo@unipg.it; 6Department of Bioscience Research, College of Dentistry, University of Tennessee Health Science Center, TN USA and The Forsyth Institute, Memphis, TN 38163, USA; fgarciagodoy@gmail.com; 7Department of Surgical Pathology, Medicine and Critical Area, University of Pisa, 56126 Pisa, Italy; gnolo78@gmail.com

**Keywords:** calcium phosphate, nano-hydroxiapatite, nanomaterials, dentine hypersensitivity, teethmate desensitizer, desensitising agents, tooth bleaching, clinical trials

## Abstract

Background: Calcium phosphate-based compounds are used to treat dental hypersensitivity (DH). Their long-term clinical behaviour needs further research. This study compared the 24-week effectiveness of Teethmate Desensitizer (TD), a pure tetracalcium phosphate (TTCP) and dicalcium phosphate dihydrate (DCPD) powder/water, to that of Dentin Desensitizer (DD), and Bite & White ExSense (BWE), both of calcium phosphate crystallites. Methods: A total of 105 subjects were selected. A random table was utilised to form three groups of 35 subjects. DH was evaluated using the evaporative sensitivity, tactile sensitivity tests, and the visual analogue scale (VAS) of pain. Response was recorded before the application of the materials (Pre-1), immediately after (Post-0), at 1 week (Post-1), 4 weeks (Post-2), 12 weeks (Post-3) and 24 weeks (Post-4). The non-parametric distribution was assessed with the Shapiro–Wilk statistical test. Intra-group differences for the six time points were evaluated with the Friedman statistical test and the Kruskal–Wallis test. Results: All the materials decreased DH after 24 weeks in comparison to Pre-1. However, the TTCP/DCPD cement showed the greatest statistical efficiency. Conclusions: The significant decrease of VAS scores produced by TD in the long term suggest the material as the most reliable in the clinical relief of DH.

## 1. Introduction

Dental hypersensitivity (DH) is a common oral disease characterised by a short and sharp sensation of pain in response to thermal, tactile, osmotic, evaporative or chemical stimuli [1,2]. As a result of differing selection criteria [3], DH has a wide prevalence rate in 3%–98% of the adult population, with a peak in 20–50 years [4]. The range includes 67% of patients suffering from transient sensitivity during bleaching treatments, whose pain negatively impacts their quality of life and leads them to stop the treatment [5,6]. In fact, the pain arising from the exposed dentinal tubules can compromise daily activities such as social interaction, eating and drinking [7]. The pain occurs due to the exposure of the cervical dentine surface following the loss of enamel or the recession of the marginal gingiva in association with the loss of cementum [8].

The most widely accepted physio-pathologic mechanism of DH is Brannström’s “hydrodynamic theory” [1,2]. According to this theory, the pain is due to fluid shifts in the oral exposed tubules as external stimuli induce an outward and/or inward flow of dentinal fluid, which indirectly stimulate the pulp nerves. Consequently, any therapy for DH has to interact with the hydrodynamic chain acting at the surface of the patient’s dentinal tubules or at the neural transmission pathway [9,10].

During the last 50 years, a large number of both self-applied and professionally administered agents have been advocated in the market for the relief of DH [11,12,13,14,15,16,17,18,19]. These materials contain a wide range of active ingredients, such as fluoride, oxalates, potassium nitrate and calcium phosphates. Calcium phosphate compounds forming biomimetic hydroxyapatite have gained considerable interest [16,20,21]. They have high biocompatibility and remineralisation capacity [22]. However, clinical studies focusing on their long-term effectiveness as dental and intra-oral agents are lacking. The materials essentially consist of calcium phosphate particles, which are able to self-set to a hard mass [23]. Tetracalcium phosphate (TTCP) and dicalcium phosphate dihydrate (DCPD) powders have largely been used to formulate calcium phosphate-containing materials due to the high solubility at neutral pH, resulting in a supersaturated solution and faster hydroxyapatite (HAP) precipitation [24]. The powder has to be properly mixed with water and be placed on the affected dentine surface for the determined time. As this procedure is crucial in achieving the best results and depends on the ability of the operator, premixed nano-HAP-containing pastes have been developed as in-office desensitisers and toothpastes [25]. In these pastes, crystals are carried in water or water-miscible liquids, sometimes with the addition of accelerators and cellulose-gelling agents to increase viscosity and the hygroscopic nature of the compound [25,26]. After application in the mouth and exposure to saliva, HAP may be precipitated and can be expected to seal the opened tubular orifices.

The aim of the present study was to assess the efficacy of a TTCP and DCPD cement in comparison to that of two HAP-containing compounds in patients suffering from DH within 24 weeks of clinical application. The null hypothesis tested was that there will be no statistical differences in DH using the three materials at the 24-week evaluation.

## 2. Materials and Methods

### 2.1. Study Design and Ethical Aspects

The study was carried out as an interventional, randomised, single-centre clinical trial. The research was ethically conducted in accordance with the Declaration of Helsinki. The protocol and informed consent forms were approved by the Ethical Committee of the University of Sassari [no. DH 2304/CE]. The study followed CONSORT guidelines and was registered at the US National Institutes of Health [ClinicalTrials.gov no. NCT02770573]. Patients were carefully informed of the study’s purpose, risks and benefits. Informed consent was obtained from all subjects prior to the study. Subjects were extensively trained for all procedures.

### 2.2. Participants

Two trained and calibrated examiners selected patients complaining about DH. Calibration of the examiners was carried out on 10 subjects prior to the trial. Duplicate examinations were carried out on 10% of the subjects during the trial. The kappa statistic was used to assess the inter-examiner reproducibility.

The medical and dental history of the patients was collected, and differential diagnosis for dental sensitivity was performed. A total of 121 subjects were assessed for their eligibility. As in a previous trial testing the ability of desensitisers [16,19], the study selection criteria were the following: (1) patients were included if they had sensitive teeth evidencing abrasion, erosion or recession with the exposure of the cervical dentine; (2) patients were excluded in case of teeth with subjective or objective evidence of carious lesions, pulpitis, restorations, premature contact, cracked enamel, active periapical infection, or after periodontal surgery or root planing up to six months. Other exclusion criteria were professional desensitising therapy during the previous three months, or use of desensitising toothpaste in the last six weeks, and exposure to drugs which could affect pain perception (e.g., antidepressants, anti-inflammatory drugs, sedatives, and muscle relaxants). There were 16 subjects who did not meet the inclusion criteria. Therefore, 105 subjects, who demonstrated hypersensitive teeth that satisfied the tactile and airblast hypersensitivity enrolment criteria described below, were included in the study.

### 2.3. Clinical Procedure

The subjects were randomly assigned to one of the three treatments (35 subjects per treatment group) as follows: Group 1—Teethmate Desensitizer (TD) (Noritake Dental Inc., Tokyo, Japan), a TTCP and DCPD calcium phosphate powder and water liquid; Group 2—Dentin Desensitizer (DD) (Ghimas, Casalecchio di Reno, Bologna, Italy), a premixed n-HAP alcohol-based gel, also containing 30% potassium oxalate; Group 3—Bite&White ExSense (BWE) (Cavex Holland, Haarlem, Netherlands), a premixed n-HAP water-based gel, also containing potassium nitrate (Figure 1).

The tested materials were used following the manufacturers’ instructions (Table 1).

A week before the experiment, subjects received oral prophylaxis treatment (scaling and polishing procedures). A non-fluoride toothpaste (Toothpaste Total Protection, Istituto Erboristico L’Angelica-Coswell, Funo di Argelato, Bologna, Italy), a soft toothbrush (Oral-B Sensitive Advantage, Procter & Gamble, Cincinnati, OH, USA) and oral hygiene instructions were also provided with the purpose of standardising habits during the study period.

The randomization process was made using a computer-generated random table using Excel software (Microsoft, Redmond, VA, USA). The level of sensitivity experienced by each patient was considered independent from the position of the hypersensitive tooth in the oral cavity. Between two and four hypersensitive teeth were assessed using validated stimuli tests (i.e., evaporative sensitivity and tactile sensitivity tests). The hypersensitive teeth were isolated with cotton rolls and stimuli were applied to each tooth. Stimuli tests were performed according to accepted procedures [27,28].

Assessment of evaporative (cold air) sensitivity: one-second application of compressed air from a triple air dental syringe at 60 (±5) psi with an operating temperature of 19 (±5) °C, perpendicular to the exposed dentine surface, from a distance of approximately 1 cm, while the adjacent teeth were isolated with cotton rolls.

Assessment of tactile sensitivity: a sharp dental explorer (EXD 11–12, Hu-Friedy, Chicago, IL, USA) was passed across the facial area of the tooth, perpendicular to its long axis, at an approximated constant force. The test was repeated three times before scores were recorded.

As previously described [15,16,19], the response to the stimuli was recorded using a visual analogue scale (VAS) (pain graded from 1 to 10). Subject’s response was recorded before the application of the material (Pre-1), immediately after (Post-0), after 1 week (Post-1), 4 weeks (Post-2), 12 weeks (Post-3), and 24 weeks (Post-4) of oral environment exposure. The same operator carried out the sensitivity test.

### 2.4. Sample Size Estimation and Statistical Analysis

The power of the study was calculated considering three groups and six repeated measures, a confidence interval of 0.95, an effect size of 0.15, an alpha of 0.05, and a drop-out rate of 20%. On the basis of those assumptions, a minimum of 35 subjects per group was requested to achieve an 85% statistical power.

A descriptive analysis was carried out for all the variables collected through an electronic ad hoc form. Qualitative and quantitative variables were summarised with percentages and medians and interquartile ranges (IQR), respectively. The non-parametric distribution was assessed with the Shapiro–Wilk statistical test.

Intra-group differences for six time points (i.e., Pre-1, Post-0, Post-1, Post-2, Post-3, and Post-4) were evaluated with the Friedman statistical test for repeated measures. Differences between groups were assessed using the Kruskal–Wallis statistical test.

A two-tailed *p*-value less than 0.05 was considered statistically significant. All statistical computations were carried out with the statistical software STATA version 14 (STATACorp LP, College Station, TX, USA).

## 3. Results

In total, 121 subjects were assessed for eligibility. However, 16 subjects were excluded because they did not meet the inclusion criteria. Therefore, 105 subjects were enrolled—56 females (mean age 43 years) and 49 males (mean age 50 years). No dropouts were recorded, and adverse reactions did not occur. Table 2 and Table 3 show median VAS scores. At baseline Pre-1 (VAS score), no significant statistical differences were observed (*p*-value of 0.11). After the administration of the materials, VAS values decreased from Post-0 to Post-4 and, overall, all three materials showed similar results using the two stimuli tests (Figure 2 and Figure 3).

### 3.1. Evaporative (Cold Air) Sensitivity

For TD, DD and BWE, intra-group differences were not statistically significant at any time point. No significant differences were found at Post-0; median (IQR) values were 0 (0–1) for TD group, 0 (0–2) for DD group and 1 (0–2) for BWE group (*p* = 0.08). However, differences were statistically significant (*p* = 0.0001) between all groups at Post-1 and Post-2 (*p* = 0.0001).

Median (IQR) values for DD were 2 [1–4], which were statistically different at Post-3 in comparison to those recorded for TD and BWE, namely 1 [0–2] and 1 [0–3], respectively (*p* = 0.007). Significant differences were found between groups at Post-4 (*p* = 0.0002) (Table 2).

### 3.2. Tactile Dentinal Hypersensitivity

Specifically, no significant differences were found at Post-0; median (IQR) values were 0 (0–1) for TD group, 0 (0–2) for DD group and 1 (0–2) for BWE group (*p* = 0.08).

Statistically significant differences between all groups were found at Post-1 and Post-2 (*p* = 0.0001). Similar differences but with lower median (IQR) values were detected at Post-3 with 0.5 (0–2) for TD, 2 (1–3) for DD and 1 (0–3) for BWE (*p* = 0.0007), as well as at Post-4 (*p* = 0.0002) (Table 3).

## 4. Discussion

The administration of the three HAP-forming agents reduced the pain-related VAS values, independently from thermal or tactile stimuli, within 24 weeks of clinical evaluation. Explaining these data, we can argue that immediately after their application, all the pastes could have stayed on the tubular surface of dentine, possibly delivering Ca-P particles into the tubules [23]. As a result, the tubular ratio and hydraulic pipeline could have been reduced, thus explaining the reduced pain and the drop of VAS values [2,9]. No significant difference in efficacy was observed between the groups at this time point. The setting phase of the pastes in oral fluids associated with the HAP crystallites deposited into the tubules can explain the behaviour of each material within the long term of this study [23]. As a consequence of the different behaviour, statistical differences were found between the agents within the time span of this experiment, even if at the last check point all of the materials had lower capacity in reducing DH values in comparison to that observed in Pre-1. Data could be interpreted following their composition and then the ability to harden on the exposed surface and resist chemical and physical challenges with stable remineralisation of the tubules [19,29,30].

TD was the most successful material among the agents tested. At Post-0, the cement significantly reduced the evaporative and tactile stimuli in comparison to Pre-1. At 1 week and 4 weeks, Post-1 and -2, respectively, TD showed higher efficacy in comparison to DD and BWE. Despite a decrease in efficiency at the 12-week control, Post-3, its performance was better when compared to DD and BWE within the 24 weeks of evaluation.

To explain the clinical behaviour of TD in DH, we can match our data to those reported in several in vitro analyses, which tested the capacity of the agent simulating the oral stress and chemical and mechanical interferences [20,23,30]. However, none of the published papers evaluated the material for more than four weeks. We can argue that the relief of DH obtained using TD can be attributed to the lower hydraulic conductance as shown in four-week in vitro experiments [20,24]. In particular, our data could support the capacity of the slurry to adhere to the exposed dentine with the formation of a layer of Ca-P components [18,22], which resulted in the occlusion of the exposed tubules, with a VAS drop immediately after the application. After one week and four weeks in artificial saliva, the Ca-P-rich layer demonstrated the presence of crystallites and crystal plugs into the dentinal tubules [20,24], being deeply integrated into the dentinal tissue, even when ultrasonication was used [18]. We could adopt the same considerations to explain the capacity of TD to firmly reduce the pain within 4 weeks of oral exposure [20,24,30]. TD has faster setting and hardening in addition to high shear resistance in the laboratory, which is related to its TTCP and DCPD composition [23]. Adequate low TTCP/DCPD ratio in the powder [31], and the presence of water as the cement liquid are key factors that drive the dissolution/reprecipitation reaction of the calcium phosphate cement in an aqueous environment [32]. As a consequence of such chemistry, TD possesses the highest degree of solubility at neutral pH, providing the greatest driving force for biological HAP formation in a highly supersaturated aqueous solution. This further explains the stable lowering of the VAS within four weeks of our clinical examination. However, the decreased effectiveness we reported after 12 weeks could be associated with a partial removal of the crystal layer leading to exposure of some tubular areas. This mechanism could explain a non-statistical increase of VAS values after 12 weeks in comparison to the previous check points. This consideration could be corroborated by in vitro observations, which reported that the TD layer could be partially affected by erosion-abrasion cycling over time [30]. However, also in this case, the exposure of dentine areas did not produce a statistically different reduction of hydraulic conductance, in comparison to non-stressed controls.

In regard to DD, a gel phase of n-HAP in alcohol carrier, it was less effective in reducing DH when compared to TD. At Post-0, DD decreased VAS median values. However, it showed a rapid decrease in effectiveness within four weeks. Conversely, at 12 and 24 weeks, DD-related VAS values showed poor variability, and were lower when compared with those recorded at Pre-1.

Similar to TD, DD decreased the pain probably due to the capacity of the slurry to stay on the affected surface of dentine and exposed tubular orifices. However, this premixed calcium phosphate cement was unsuccessful, probably following the impact of the alcohol carrier on the viscosity of the slurry [26]. Furthermore, a high reactivity with moisture and slow hardening reported in premixed compounds [23] could have caused an easy detachment from the surface of dentine, explaining the VAS increase within the four-week exposure. In addition, the stability of values recorded at 12 and 24 weeks could further support the long time span required by the material to form firm crystal tubular plugs in salivary fluids [30].

BWE, a gel phase of n-HAP in a water vehicle, became less effective immediately after application, Post-0, in comparison to the other two materials. However, at Post-1 and -2 BWE decreased DH and the VAS values, which again increased within 12–24 weeks. According to the manufacturer, BWE is a blend of crystalline n-HAP in a hydrodispersing clay to accelerate the dispersion and penetration of HAP into the dentinal tubules. Indeed, we can argue that BWE had the capacity to spread and stay on the surface of dentine, thus reducing the dentine tubule diameter and the sensitivity of the nerve [3]. Furthermore, we can speculate that the rate of tubular occlusion increased within four weeks of exposure in an aqueous environment [30]. This consideration is sustained by the ability of n-HAP toothpastes to occlude dentinal tubules with a plugging rate >90% by means of Ca-P ions over time [33,34,35,36]. Nevertheless, after 12 weeks and within 24 weeks the decreased effectiveness could be associated with a low resistance of the formed crystallites against chemical and physical oral challenges.

## 5. Conclusions

All the HAP-forming materials decreased DH during the follow-up of 24 weeks. However, statistically significant differences between the materials were found. Thus, the first null hypothesis was rejected.

The better performance of TD in comparison to DD and BWE can be attributed to its formulation, which may have allowed faster formation of HAP crystallites and the highest rate of resistance of tubular occlusions.

## Figures and Tables

**Figure 1 nanomaterials-09-01748-f001:**
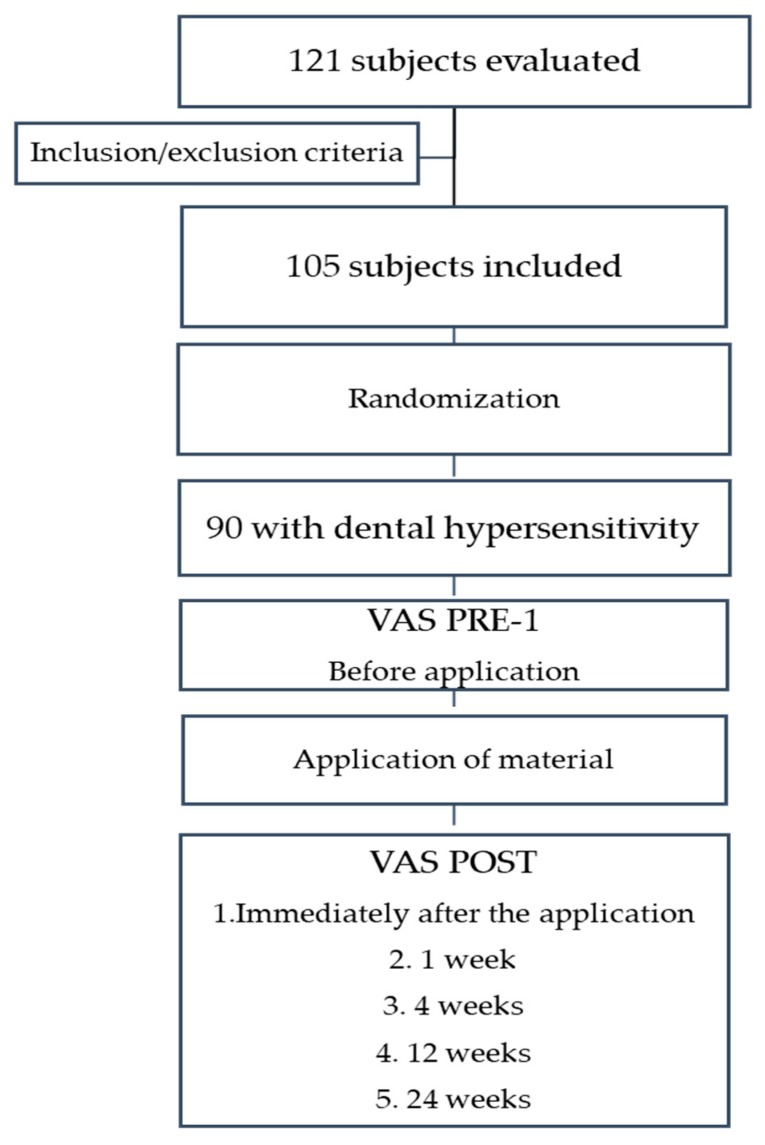
Flowchart of the study. VAS, visual analogue scale.

**Figure 2 nanomaterials-09-01748-f002:**
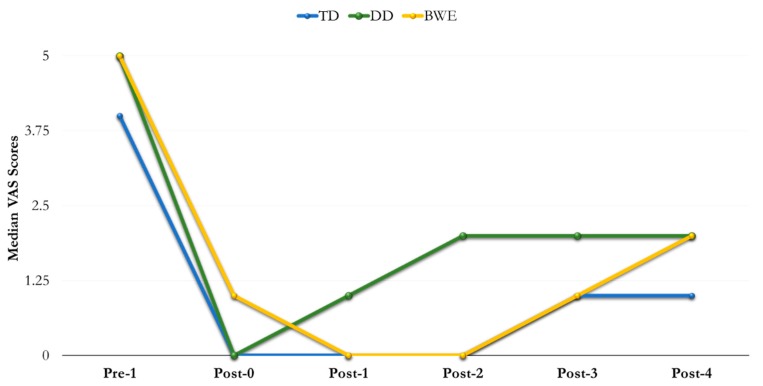
Trend of visual analogue scale (VAS) scores recorded by evaporative stimulus.

**Figure 3 nanomaterials-09-01748-f003:**
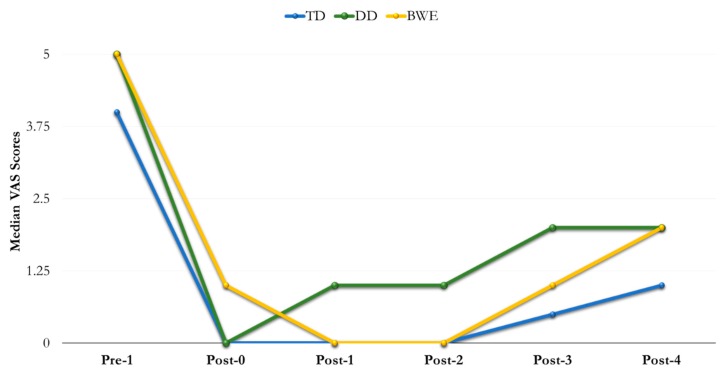
Trend of VAS Scores recorded by tactile stimulus.

**Table 1 nanomaterials-09-01748-t001:** Composition and application mode of the desensitising agents (manufacturer’s data).

Material	Manufacturer	Main Components	Batch No.	Mode of Application
Teethmate™ Desensitizer (TD)	Kuraray Noritake Dental Inc., Tokyo, Japan	Powder: Tetra-calcium phosphate, dicalcium phosphate anhydrous. Liquid: Water, preservative	041,118	1. Mix the powder with the liquid for 30 s, then rub the obtained slurry on the dried affected dentine for 30 s. Rinse the excess of the slurry with water spray or by having the patient rinse.
Dentin Desensitizer (DD)	Ghimas, Casalecchio di Reno, Bologna, Italy	A proprietary gel phase of n-HAP in an alcohol vehicle	2015-001	1. After cleaning, apply the paste on the saliva-wetted dentine surface using a brush for 45 s. 2. Repeat the procedure for at least 3 times; 3. Rinse with water spray.
Bite&White ExSense (BWE)	Cavex Holland, Haarlem, Netherlands	A proprietary gel phase of nano-HAP in a water vehicle	150,702	1. Dispense a small amount of the paste onto a clean finger. Gently apply the gel on all the surfaces of the tooth allowing to remain for 10 min. 2. Spit out any excess of the gel and rinse the mouth with water.

**Table 2 nanomaterials-09-01748-t002:** Comparison of VAS scores by evaporative (cold air) stimulus during different time points.

	TD	DD	BWE	*p*-Value
Pre-1, median (IQR)	4 (2–6)	5 (4–7)	5 (2–6)	0.01 ^1^
Post-0, median (IQR)	0 (0–1)	0 (0–2)	1 (0–2)	0.08
Post-1, median (IQR)	0 (0–0)	1 (0–3)	0 (0–0)	0.0001 ^2^
Post-2, median (IQR)	0 (0–2)	2 (0–3)	0 (0–0)	0.0001 ^3^
Post-3, median (IQR)	1 (0–2)	2 (1–4)	1 (0–3)	0.0007 ^4^
Post-4, median (IQR)	1 (0–2)	2 (1–5)	2 (2–4)	0.0002 ^5^
*p*-value	<0.0001	<0.0001	<0.0001	

^1^ DD vs. BWE *p*-value = 0.008; TD vs. DD *p*-value = 0.003; ^2^ DD vs. BWE *p*-value < 0.0001; TD vs. BWE *p*-value = 0.002; TD vs. DD *p*-value < 0.0001; ^3^ DD vs. BWE *p*-value < 0.0001; TD vs. BWE *p*-value < 0.0001; TD vs. DD *p*-value < 0.0001; ^4^ DD vs. BWE *p*-value = 0.001; TD vs. DD *p*-value = 0.0002; ^5^ TD vs. BWE *p*-value = 0.0002; TD vs. DD *p*-value = 0.0001.

**Table 3 nanomaterials-09-01748-t003:** Comparison of VAS scores by tactile stimulus during different time points.

	TD	DD	BWE	*p*-Value
Pre-1, median (IQR)	4 (2–6)	5 (4–7)	5 (2–6)	0.01 ^1^
Post-0, median (IQR)	0 (0–1)	0 (0–2)	1 (0–2)	0.08
Post-1, median (IQR)	0 (0–0)	1 (0–2)	0 (0–0)	0.0001 ^2^
Post-2, median (IQR)	0 (0–2)	1 (0–3)	0 (0–0)	0.0001 ^3^
Post-3, median (IQR)	0.5 (0–2)	2 (1–3)	1 (0–2)	0.0007 ^4^
Post-4, median (IQR)	1 (0–2)	2 (1–5)	2 (1–3)	0.0002 ^5^
*p*-value	<0.0001	<0.0001	<0.0001	-

^1^ DD vs. BWE *p*-value = 0.008; TD vs. DD *p*-value = 0.003; ^2^ DD vs. BWE *p*-value < 0.0001; TD vs. BWE *p*-value = 0.002; TD vs. DD *p*-value < 0.0001; ^3^ DD vs. BWE *p*-value < 0.0001; TD vs. BWE *p*-value < 0.0001; TD vs. DD *p*-value < 0.0001; ^4^ DD vs. BWE *p*-value = 0.001; TD vs. DD *p*-value = 0.0002; ^5^ TD vs. BWE *p*-value = 0.0002; TD vs. DD *p*-value = 0.0001.

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
