# Peer review of "Effectiveness of Calcium Phosphate Desensitising Agents in Dental Hypersensitivity Over 24 Weeks of Clinical Evaluation"

_nanomaterials, 2019, doi:10.3390/nano9121748_

Round 1

Reviewer 1 Report

Nice study. Figure 1 and 2 should be redone in 2D instead. Very difficult to read in present format. 

Author Response

Dear Reviwer,

first of all, many thanks for appreciating our work. 

Regarding your right observation about the figures 1 and 2, we have changed them in a 2D graph.  As you can see in the new version of the manuscript, the results are now improved thanks to your suggestion.

Reviewer 2 Report

There were a little reports to evaluate long term clinical behavior of calcium-phosphate compound for the treatment of DH. This study was a long term study, 24 weeks of clinical evaluation. This study was also well-designed and carried out as an interventional, randomized, single-center clinical trial. It has discovered novel scientific findings that fasten formation of HAP crystallites and the highest rate of resistance of tubular occlusions are important for the relief of DH symptoms.   

Author Response

Dear Reviewer,

thanks a lot for appreciating our work

Reviewer 3 Report

I read, with interest, the author's study on the effectiveness of calcium-phosphate desensitizing agents in this clinical evaluation. The manuscript was well-crafted with sound methods and a clear conclusion, based on the appropriate study. Other than grammar and word choice issues, I really have no strong criticisms of this work. The conclusions have potentially interesting and important ramifications and helps to elucidate the role of dentin tubule occlusion via hydroxyapetite. I think the "broad interest" is of sufficient novelty to warrant publication. I have a few minor things to comments on:

1) In the introduction, just point out that the 3-98% large range was the result of differing selection criteria.

2) Figure A1 and A2 would be better on a 2D graph. These 3-D things are hard for comparison purposes.

3) Have this (quickly) edited by a native speaker. There were some area where it was just awkward but other areas where it might actually affect the meaning of your intentions.

Author Response

Dear Reviewer,

First of all, many thanks for your kind comments and suggestions. 

1) in the introduction: we pointed out the wide range in percent of DH is the result of the methods adopted in the different studies. You can see the changes in red in the template.

2) Figures A1 and A2 were changed in a 2D graph. Also, the new figures were inserted in a new graphical abstract that substitutes the older one.

3) the manuscript was checked in the English form by a mother tongue, also one of the authors of the manuscript. One sentence in the introduction section was changed and it is in red in the new template. However, we wonder if you  should indicate the parts in the text you like to be changed.